# Effect of Compression Stockings after Endovenous Laser Ablation of the Great Saphenous Vein with a 1470 nm Diode Laser Device and a 2ring Fiber

**DOI:** 10.3390/jcm10173861

**Published:** 2021-08-27

**Authors:** Laura Fischer, Uldis Maurins, Eberhard Rabe, Juris Rits, Arnolds Kadiss, Sandra Prave, Rets Vigants, Felizitas Pannier

**Affiliations:** 1Department of Dermatology, University of Essen, Esmarchstraße 14, 45147 Essen, Germany; 2Vein Clinic, LV-1000 Riga, Latvia; uldis.maurins@flebomedika.lv (U.M.); juris.rits@flebomedika.lv (J.R.); arnolds.kadiss@flebomedika.lv (A.K.); sandra.prave@flebomedika.lv (S.P.); rets.vigants@flebomedika.lv (R.V.); 3Department of Dermatology, University of Bonn, Sigmund-Freud-Str. 25, 53127 Bonn, Germany; Eberhard.rabe@ukbonn.de; 4Private Practice Dermatology & Phlebology, Helmholtzstr. 4, 53123 Bonn, Germany; info@dr-pannier.de; 5Department of Dermatology, University of Cologne, Kerpener Str. 62, 50937 Cologne, Germany

**Keywords:** endovenous thermal ablation, 2ring fiber, great saphenous vein, compression, radial fiber

## Abstract

The aim of this study was to demonstrate the effects of compression following the endovenous laser ablation (EVLA) of incompetent great saphenous veins (GSVs) using a 1470 nm diode laser (Ceralas E 1470 nm, biolitec) and a 2ring radial fiber (ELVeS Radial 2ring™, biolitec). In this single-center prospective study, 150 legs of 150 consecutive patients were randomly allocated to one of three groups (A, B, and C). Group A patients did not undergo postoperative compression. Group B patients wore a thigh-length graduated compression stocking (23–32 mmHg) for 7 days, whereas group C patients wore the same stocking for 28 days. No additional phlebectomies or sclerotherapies were performed. Investigations were performed prior to intervention, at the day of intervention (D0), at day 7 (D7), and at day 28 post intervention (D28). The primary endpoint was post-interventional pain measured on a 10-point scale. A significant but small pain decrease was observed in the first week of compression, by comparing group B’s mean pain scores to those of group A (*p* = 0.009). Wearing a compression stocking after EVLA reduced pain within the first week on a significant, but low level. Taking the very low differences in pain levels into account, the difference may not be clinically relevant and post-treatment compression may not be necessary if no additional phlebectomies or sclerotherapies are performed.

## 1. Introduction

Endovenous laser ablation (EVLA) is an effective method of treating incompetent great saphenous veins (GSV) [1,2,3], with occlusion rates demonstrated to reach approximately 95% after five years [4]. Compression treatment with bandages or compression stockings is used routinely after saphenous ablation to reduce pain, bruising and other side effects [5,6]. Recent studies demonstrated that the main effect of compression following GSV interventions is pain reduction within the first 7 days after treatment [7,8,9,10,11,12,13,14,15]. In most of these studies, additional phlebectomies were performed. The aim of this study was to demonstrate the outcomes and side effects of compression therapy four weeks after EVLA of the GSV using a 1470 nm diode laser (Ceralas E 1470 nm, biolitec) and a 2ring radial fiber (ELVeS Radial 2ring™, biolitec) with and without compression stockings. No additional phlebectomies or sclerotherapy of varicose veins were performed.

## 2. Materials and Methods

In this single-centre prospective study, 150 legs in 150 consecutively sampled patients from the Dr. Maurins Vein Clinic in Riga, Latvia, treated by EVLA for GSV incompetence between November 2011 and March 2013, were randomly assigned to one of three groups: Group A underwent no postoperative compression; Group B used postoperative compression with a thigh-length graduated compression stocking (23–32 mmHg) for 7 days; Group C used the same kind of stocking for 28 days postoperatively. In groups B and C, initial eccentric compression with cottonwool was performed following the procedure. The stockings were applied every morning until evening. A sample size of 50 patients per group was calculated at 80% power and a 5% significance level, assuming a 20% drop out rate and an equal number of patients per group. To facilitate randomization, the consecutively ordered patients were assigned to groups A, B or C in an alternating fashion until 50 patients were included in each group.

All patients were examined clinically and by duplex ultrasound by an experienced phlebologist prior to intervention (screening visit), at the day of the intervention (D0) and at the follow-up visits after the procedure at day 7 (D7) and day 28 (D28) to assess side effects, complications and occlusion of the treated vein. 

Duplex ultrasounds were performed in an upright position. Reflux was defined as retrograde flow of >0.5 s duration after a Valsalva maneuver or manual compression and decompression of the distal vein. All treated veins were assessed pre- and postoperatively and even a slight marginal flow or reflux in a largely occluded vein was assessed as not occluded. The entire deep venous system of the legs was checked for DVT. 

The main outcome parameter was patient-reported postoperative pain. Secondary outcome parameters were: the improvement of the rVCSS score, the use of analgesics, DVT, ecchymoses, bleeding and other adverse events, the occlusion rate of GSV, leg circumference, time off work and normal activity, and patient satisfaction.

Clinical evaluations involved clinical classification (C of CEAP) and the revised venous clinical severity score (rVCSS) [16,17]. The rVCSS has 10 questions with 0–3 points for each item, yielding a minimum score of 0 points and a maximum of 30. Patients reported pain was assessed on an 11-point numerical scale ranging from no pain at all (0) to worst pain (10) in a diary. Patients’ satisfaction with their treatment was assessed using 5-point-scales ranging from 0 to 4. The questions were: “Are you satisfied with the method being used?” (0 = very satisfied, 1 = satisfied, 2 = fairly satisfied, 3 = not satisfied, 4 = extremely unsatisfied), and “would you choose endovenous laser therapy again?” (0 = definitely, 1 = probably, 2 = don’t know, 3 = probably not, 4 = definitely not). To assess for possible leg swelling after the procedure, the circumference of the lower leg was measured on D0, D7 and D28 at three different locations: the smallest ankle circumference, the mid-calf, and below the knee.

EVLAs were performed with a 1470 nm Diode laser (Ceralas E, biolitec). The entire procedure was performed under duplex guidance (Imagic Agile, Kontron MEDICAL) using cold [18] (5 °C) tumescent local anesthesia with 0.05% lidocaine. A similar volume of tumescence fluid was used in all groups. No additional treatment, such as phlebectomies or sclerotherapies for incompetent tributaries, were performed in the same session or during follow up. 

GSVs were accessed at the most distal insufficient points using a 17-gauge needle. The 600 μm radial 2ring fiber was introduced through a micro puncture set and the tip was positioned at the level of the GSVs’ terminal valve under duplex guidance. The tumescent local anesthesia was then applied perivenously under duplex guidance. Laser treatment was carried out in a continuous mode with a power of 10 W. Prophylactic anticoagulation was given for 7 days to all patients. The patients were mobilized immediately after the intervention. The NSAID ibuprofen, 200 mg, was prescribed in cases of postoperative pain.

The results are reported as absolute numbers, percentages (%), mean values, and standard deviations (SDs). Two-sided t-tests were used to test the hypothesis that differences will occur between the treatment groups. *p*-values below 5% were considered significant. 

## 3. Results

The patients’ general and technical data are presented in Table 1. No significant differences could be found for the general data between the groups except for the distribution of C-stages. 

Follow-up was 28 days. Four patients (2.7%) were unavailable for follow up, two in group A and 2 in group B. The average linear endovenous energy density (LEED) was 61 (42–104) J/cm in group A, 62 (36–94) in group B, and 64 (31–122) in group C. The average endovenous fluence equivalent (EFE) was 31 (20–43) J/cm^2^ vein in group A, 31 (19–45) in group B, and 31 (20–46) in group C. The results are shown in Table 2. 

All GSVs showed complete occlusion without the early recurrence of reflux within 28 days. The pain score in group A reduced from a mean of 1.4 on the day of the intervention, to a mean of 0.9 at days 1–7, and to 0.5 at days 8–28. In groups B and C, the corresponding values were 1.0, 0.4, 0.4 and 1.5, 0.6, 0.4 respectively. A total of 83% (group A), 79% (group B), and 86% (group C) of patients did not develop any postoperative pain and 88%, 88% and 92%, respectively, did not use any analgesics. The VCSS dropped significantly from 6.4 before the procedure to 3.2 at D28 in group A, 5.6 to 2.6 in group B, and 6.1 to 2.9 in group C. The patients returned to normal daily activities after an average of 0.4, 0.5, and 0.4 days in groups A–C, respectively. No severe complications such as deep venous thrombosis, pulmonary embolism, skin burns, motor nerve lesions, or the formation of arterio-venous fistula occurred in any of the treated legs. In seven group A patients, two group B patients, and five group C patients, local paresthesia occurred. Hyperpigmentation developed in seven group A patients, one group B patient, and three group C patients. Two group A patients, three group B patients, and one group C patient developed phlebitis. One patient in group B developed minor bleeding at the puncture site. Postoperative ecchymoses in the tracks of treated GSVs developed in 10 patients in group A, 11 patients in group B, and 8 patients in group C prior to D7. At D28, remaining ecchymoses were only present in two patients in group A and in one group B patient.

## 4. Discussion

The results of this study show that wearing compression stockings after endovenous laser ablation (EVLA) of the great saphenous vein (GSV) can reduce pain in the first week after the intervention significantly but on a low pain level. There is no further benefit after one week and no influence on other adverse events or on the short-term occlusion rate.

EVLA is an effective method of treating incompetent saphenous veins [1,2,3]. The development of modified fiber tips that reduce the high energy level concentrated at a single point has reduced post-treatment pain and bruising [19,20]. The radial 2ring fiber is a further step in this direction [21]. In our study, we demonstrated excellent early occlusion rates and a very low level of post-treatment pain. rVCSS scores improved significantly. Long-term follow-up studies would be necessary to evaluate long-term occlusion rates. Most of our patients returned to normal daily activities the same or the next day, which is in accordance with other studies [11,12,13,14,15].

Compression bandages or compression stockings following GSV treatment are recommended in most of the available guidelines and consensus documents [6]. However, the evidence for improved outcomes remain poor [5]. In a metanalysis, Huang et al. demonstrated that no evidence exists for the benefit of long-term compression after varicose vein surgery [7]. Pain was only significantly reduced during the first week of compression therapy After 4 and 6 weeks, no differences existed between the compression and non-compression groups [7]. In a prospective randomized study, Houtermans-Auckel treated patients who underwent varicose vein stripping with compression bandages for 3 days and then randomly assigned patients into either a group that wore compression stockings (23–32 mmHg) for four weeks, or a group that did not continue with further compression therapy [8]. The compression group showed a small but significant decrease in leg volume compared to the control group. No other additional benefit in the compression group was found. Reich-Schupke et al. compared the effects of low-pressure (18–21 mmHg) thigh-high compression stockings with high-pressure (23–32 mmHg) thigh-high compression stockings for six weeks following varicose vein surgery [9]. They found a significant improvement in the resolution of oedemas and feelings of tightness after one week. After 6 weeks, no significant differences had been found. Hamel-Desnos et al. compared the results of foam sclerotherapy in saphenous varicose veins without and with compression stockings (15–20 mmHg) for 3 weeks [10]. They did not find significant differences in occlusion rates, symptoms, or quality of life assessment scores after 14 and 28 days. However, evaluation was not performed in the first week and the pressure of the stocking was lower than in comparable studies. Concerning the EVLA of GSVs, Marcia Lugli and colleagues compared 100 consecutive patients without and 100 patients with an eccentric compression device on their treated GSVs [11]. One week after the procedure, the compression group demonstrated significantly less postoperative pain compared to the non-compression group. On a scale ranging from 0 (no pain) to 10 (worst pain), the mean value in the compression group was 1.4 and in the control group, it was 4.9 (*p* < 0.001). In the compression group, significantly less ecchymoses appeared. A 940 nm diode, 30-Watt laser and a bare fiber were used. In a recent study, Bakker et al. compared the effects of wearing compression stockings after EVLA of GSVs for 48 hours in group A with 7 days in group B [12]. After one week, the patients of group A reported significantly more pain (VAS score 3.7 vs. 2.0) and physical dysfunction when compared to group B. No differences were found after 6 weeks. The occlusion rate was 100% in both groups. Additionally, in this study, a bare fiber was used in combination with an 810 nm diode laser. Bootun et al. reported a significantly lower median pain score after the EVLA of saphenous veins in a compression group on days 2–5, compared to a no compression group in the COMETA trial [13]. Patients with additional phlebectomies and compression stockings had significantly lower pain scores on days 1–3, day 5, and day 7 compared to those who did not undergo compression [13]. No significant differences concerning occlusion rates, quality of life, or bruising were found [13]. Elderman et al. demonstrated a small but significant reduction in postoperative pain and the use of analgesics when wearing compression stockings for 2 weeks after EVLT for great saphenous vein insufficiency compared with not wearing compression stockings [14]. In the initial 24-h period, compression bandages were applied to all patients [14]. Ye et al. randomized a total of 400 patients into two groups: one with, and one without compression following EVLA of the great saphenous vein [15]. In the first week, patients with compression experienced less pain (*p* < 0.001) and oedema (*p* = 0.01). There were no significant differences in the quality of life or in the time taken to return to work [15]. Onwudike et al. compared the effect of compression or no compression on the occlusion rate of incompetent saphenous veins treated by radiofrequency ablation (RFA) at 12 weeks and found no difference [22]. There was also no statistically significant difference in AVSS and rVCSS [22].

Patient-reported evaluation of pain is the gold standard in acute and chronic pain [23]. In many cases, unidimensional scales such as Visual Analogue Scales (VASs) or numerical rating scales (NRSs) are used with a good comparability and reliability of the results in adults [23]. However, these approaches are limited due to social, cognitive, or contextual influences [24]. Independently of the pain caused by the acute treatment, any patient may start the VAS or NRS evaluation with an individual level of pre-treatment pain which may even not be due to the treated disease [24]. In addition, pain severity may be rated differently on an individual basis. “Scores may be meaningful within patients over time but not necessarily across patients” [24]. The clinical relevance of differences in pain levels is controversial. Although no international consensus exists, a difference of 30%, or two points, may represent a clinically important difference [24]. 

In our study, we demonstrated a significantly lower pain score in the first week of compression therapy, by comparing the scores of group B (7 days of compression) to group A (no compression). This is in line with the results of other studies reviewed here [11,12,13,14,15]. The overall pain levels of patients in our study were very low. On the scale from 0 to 10 (worst pain), the pain levels in the first postoperative week were only 0.9 for group A and 0.4 for group B (*p* = 0.009). Although this difference is statistically significant, a difference of 0.5 points on a scale between 0 and 10 may not be considered clinically relevant. The pain level in group C was 0.6. This difference was not statistically significant (*p* = 0.07). Only six patients in group A and B, respectively, and four patients in group C took any analgesics. It also must be considered that the mean pain score on the operation day was in the range of 1.0 to 1.5, and that this dropped significantly in all groups, even those without compression. We could not demonstrate significant differences between the treatment groups concerning rVCSS, side effects, return to normal activity or work, patient satisfaction, and leg circumferences after 1 and 4 weeks. No increased leg circumferences, indicating oedema, after EVLA could be demonstrated. Conversely, the circumferences dropped slightly after the procedure in all groups without reaching significance. Compared to the study of Lugli, no eccentric compression using pads along the treated vein was used [11]. A thigh-high graduated compression stocking with an ankle pressure of 23–32 mmHg produces a pressure at the thigh which is in the range of only 10–15 mmHg, due to the degressive pressure of the stocking and the larger radius of the leg at that region (Laplace’s law). This may explain the low pain score differences between the study groups. The overall low pain level may be because no additional phlebectomies were performed. Another explanation is the use of a modern EVLA device with a 2ring radial fiber, which may cause less trauma in the surrounding tissues of the treated vein and less vein perforations compared to bare fibers [4,20]. The relatively high rate of bruising in the first week may be caused not only by EVLA itself but also by the injections for tumescence anesthesia.

## 5. Conclusions

Wearing a graduated compression stocking after the EVLA of GSVs in patients without additional phlebectomies or sclerotherapy significantly reduces pain within the first week. Taking the very low pain levels into account, the pain reduction, though statistically significant, may not be clinically relevant. Post-ablation compression seems to have no influence on occlusion rates or post-intervention rVCSS improvement in short-term observation.

## Figures and Tables

**Table 1 jcm-10-03861-t001:** Patients’ general and technical treatment data (GSV = great saphenous vein, *n* = number, BMI = body mass index, C = clinical class CEAP classification, LEED = linear endovenous energy density, EFE = endovenous fluence equivalent, *p* = *p*-value, ns = difference not significant).

Parameter	Group A	Group B	Group C	*p*
Patients at inclusion (*n*)	50	50	50	ns
Patients at follow-up (*n*)	48	48	50	ns
Female (%)	88	83	82	ns
Mean age (years) (range)	51 (24–78)	49 (25–73)	52 (24–79)	ns
BMI (kg/cm^2^) mean (range)	27 (19–45)	28 (18–39)	27 (19–44)	ns
C of CEAP				<0.05
C2 (CEAP)	17	19	20	
C3 (CEAP)	8	14	12	
C4 (CEAP)	22	15	17	
C5 (CEAP)	1	0	1	
GSV diameter (mm) at 3 cm, mean (range)	8.1 (3–15)	7.6 (4–14)	8.3 (3–21)	ns
Treated length GSV (cm) mean (range)	61 (43–88)	56 (30–83)	59 (32–88)	ns
LEED (J/cm) mean (range)	61 (42–104)	62 (36–94)	64 (31–122)	ns
EFE (J/cm^2^) mean (range)	31 (20–43)	31 (19–45)	31 (20–46)	ns

**Table 2 jcm-10-03861-t002:** Clinical results after EVLA (*n* = number of participants, D0 = day of intervention, D1–D7 = follow-up of the first week, D8–D28 = follow-up of week 2–4, *p* = *p*-value, analgesics = number of patients who took analgesics, rVCSS = venous clinical severity score, occlusion = occlusion rate, normal activity = return to normal activity, work = return to work, satisfaction = satisfied with method (score 1–4), repeat = would choose method again (score 1–4).

Parameter	Group A (*n* = 48)			Group B (*n* = 48)			Group C (*n* = 50)			*p*
Follow-up	D0	D1–D7	D8–D28	D0	D1–D7	D8–28	D0	D1–D7	D8–28	
Pain score, mean (SD)	1.4 (1.4)	0.9 (0.9)	0.5 (0.9)	1.0 (1.4)	0.4 (0.7)	0.4 (0.7)	1.5 (1.9)	0.6 (0.8)	0.4 (0.7)	*
Analgesics D1–D28, (*n*)	6			6			4			ns
Follow-up	D0	D28		D0	D28		D0	D28		
rVCSS, mean (SD)	6.4 (4.2)	3.2 (3.0)		5.6 (3.6)	2.6 (2.7)		6.1 (3.8)	2.9 (3.0)		**
Occlusion rate (%)	100			100			100			
Adverse events										ns
DVT (*n*)	0			0			0			
Infection (*n*)	0			0			0			
paresthesia (*n*)	7			2			5			
Pigmentation (*n*)	7			1			3			
Phlebitis (*n*)	2			3			1			
Bleeding (*n*)	0			1			0			
Ecchymoses (*n*) D7	10			11			8			
Return to normal activity, days, mean (SD)	0.4 (1.1)			0.5 (0.7)			0.4 (0.5)			ns
Return to work, days, mean (SD)	0.8 (1.3)			1.0 (1.9)			0.9 (1.4)			ns
Satisfaction, mean (SD)	0.5 (0.7)			0.4 (0.5)			0.4 (0.6)			ns
Reiterate, mean (SD)	0.4 (0.6)			0.3(0.5)			0.3 (0.6)			ns
Circumference (cm)	D0	D7	D28	D0	D7	D28	D0	D7	D28	ns
Ankle mean, (SD)	23.5 (1.8)	23.1 (1.9)	23.2 (1.8)	24.2 (2.7)	23.7 (2.4)	23.8 (2.5)	24.0 (2.0)	23.3 (1.8)	23.4 (1.9)	
Calf mean, (SD)	39.3 (3.7)	38.6 (3.8)	38.7 (3.8)	39.8 (3.9)	39.1 (3.9)	39.2 (4.0)	39.6 (3.7)	38.6 (3.7)	38.7 (3.8)	
Below knee mean, (SD)	36.4 (3.4)	35.8 (3.2)	36.0 (3.3)	36.9 (4.0)	36.3 (4.1)	36.7 (4.2)	36.7 (3.0)	35.8 (3.1)	36.0 (3.0)	

* Difference between D0 in group A, B and C not significant (ns), difference between D8–D28 in A, B and C ns, significant difference between A and B for D1–D7 (*p* = 0.009), difference between A and C for D1–D7 = 0.07, significant difference between D0 and D1–D7, D8–28 in all groups (*p* < 0.01). ** significant difference between D0 and D28 in all groups (*p* < 0.05), no significant difference between groups.

## Data Availability

The data presented in this study are available on request from the corresponding author. The data are not publicly available due to private storage regulations.

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
