# Peer review of "Effect of Compression Stockings after Endovenous Laser Ablation of the Great Saphenous Vein with a 1470 nm Diode Laser Device and a 2ring Fiber"

_jcm, 2021, doi:10.3390/jcm10173861_

Round 1

Reviewer 1 Report

In the current study, the authors investigated the effect of compression after endovenous laser ablation of incompetent great saphenous veins. The single-center prospective study included a nice number of 150 consecutive patients, and randomized these into three groups,  group A: without postoperative compression; group B: with a thigh-length graduated compression stocking for 7 days, and group C, thigh-length graduated compression stocking for 28 days.

The primary endpoint was post-interventional pain measured on a 10-point-scale. They showed that there was a significant reduction of pain in the first week with compression in group B compared to no compression in group A. In general the study is well conducted and the findings are clearly presented, but some questions and remain:

Major

  • Why was the 10-point-scale pain measurement used? It seems that the 10-point-scale was not fit to measure the range of pain observed.

Was a different scale score better ? Or is pain actually not a real clinical problem in these patients ?

The absolute pain reduction is now very low, even though a very large (>50%) relative reduction is noticed in group B compared to group A.

  • JCM recommends following the CONSORT guidelines, which suggest to use registration of prospective clinical trials. Why was this study not registered?

Author Response

Dear reviewer,

thank you very much for your suggestions which I answer in the following:

  1. Why was the 10-point-scale pain measurement used? It seems that the 10-point-scale was not fit to measure the range of pain observed. Was a different scale score better ? Or is pain actually not a real clinical problem in these patients ? Answer: We use this 10-point-scale pain measurement routinely in the follow-up evaluation of all our patients. Therefore we used it also in this study. A VAS for pain would have been an alternative.
  2. The absolute pain reduction is now very low, even though a very large (>50%) relative reduction is noticed in group B compared to group A. Answer: This is true but also a VAS evaluation would have given low values.
  3. JCM recommends following the CONSORT guidelines, which suggest to use registration of prospective clinical trials. Why was this study not registered? Answer: At the time when the study was planned a longer time ago we where not aware of this recommendation.

Reviewer 2 Report

  1. The widely accepted rationale to use the compression after the treatment of venous insufficiency is minimizing superficial and deep vein thrombosis, inflammation, pain, bruising, bleeding and hematoma, thus shortening recovery time. However, among aforementioned the pain is the most difficult parameter to assess, since the pain assessment is always semiquantitative, as we do not have any objective method for its measurement. Moreover, it is subjective and strongly influenced by other conditions, including patient psychophysical condition or any analgesics used. In fact, the use of tumescent anesthesia significantly affects the pain perception after procedure, and this effect, depending on the composition of used solution, may persist up to several hours (or even longer). Noteworthy, in present study the important factor should be the volume of solution used in tumescent anesthesia and Authors should at least shortly comment that issue. If the tumescence was similar in all groups, did Authors really expect any significant difference in pain score among them at Day 0?
  2. Several studies have shown the significant benefit from short-term compression and, indeed, no further advantage from its prolonged use (compare Ann Vas Dis 2021;14(2):122–131.; or short review by Mosti in Phlebology 2013;28 Suppl 1:21–24.). However, according to Laplace’s law the interstitial pressure provided by compression stockings alone on the thigh level is far ineffective to influence superficial system. Therefore, to achieve adequate local pressure in the treated area the use of eccentric compression is required. Although this issue was mentioned in discussion, Authors should shortly explain, why they decided to use regular stockings instead of the recommended eccentric compression. Thus, it seems logical that, at least in that aspect, it shouldn't be any difference between the use of regular compression stockings or their lack.
  3. The effectiveness of any method, assessed within 4 weeks after the treatment of venous insufficiency has no clinical relevance, since the failure in short-term observation usually results from technical issues during the procedure. Since the study was performed almost 8-9 years ago, it would be exciting to know its results from longer follow-up, especially the long-term results in patients with large diameter veins (>15 mm). Authors should at least shortly comment that issue.
  4. There is some inconsistency in the description of study methodology. The vein assessment was done at the levels of 3, 25 and 50 cm from sapheno-femoral junction. Which of these measurements was used for comparison? From practical point of view, it is very important for the outcome (especially, the risk of recanalization), whether the vein diameter reaches 21 mm at sapheno-femoral junction, or on the level of middle thigh. Furthermore, in the table 1. the length of treated vein in some patients is less than 50 cm (from 30 to 88 cm). What about the interpretation of the assessment results on the level of 50 cm from SFJ in those cases? Authors should shortly comment that issue.
  5. Methods (line 86): The normal ELVeS Radial 2 ring™ fiber is 1.8 mm, whereas the thinner one (Slim) is 1.2 mm in diameter. Please, verify, since to my best knowledge there is no 600 μm fiber available in Biolitec company.
  6. Two final sentences at the end of Abstract and in Conclusions are inconsistent. At first Authors have concluded that "Wearing a compression stocking after EVLA reduces pain within the first week on a significant, but low level." and then "Taking the very low differences in pain levels in account, the difference may not be clinically relevant and post-treatment compression may not be mandatory if no additional phlebectomies or sclerotherapies are performed." What is the rationale to postulate, that EVLA with sclerotherapy does require the use of compression, whereas EVLA alone does not? Authors did not include any auxiliary treatment in their analysis. Moreover, the recommendations for the use of compression after sclerotherapy are similarly ambiguous and lacking the good evidence as for EVLA. Authors should better explain their point of view and possibly should refer to recent papers by Pihlaja et al. - Eur J Vasc Endovasc Surg. 2020 Jan;59(1):73-80; Tan et al. - J Vasc Surg Venous Lymph Disord 2021;9(1):264-274, and also European Guidelines by Rabe et al. from Phlebology 2013. Furthermore, the last sentence in conclusions should be accompanied by addition “in short-term observation”, since no long-term results have been presented. 

Author Response

Dear reviewer, thank you very much for your very helpful comments and suggestions. In the following please find my answers point by point:

  1. Noteworthy, in present study the important factor should be the volume of solution used in tumescent anesthesia and Authors should at least shortly comment that issue. If the tumescence was similar in all groups, did Authors really expect any significant difference in pain score among them at Day 0? Answer: We agree that the tumescence anesthesia is a very important factor to reduce pain after the procedure. We used a comparable (no significant difference) amount of fluid in all groups and we would not expect a significant pain difference at day 0. However the pain intensity during the following days should no longer be influenced by the tumescence fluid amount. We added a sentence in the text.
  2. Authors should shortly explain, why they decided to use regular stockings instead of the recommended eccentric compression. Thus, it seems logical that, at least in that aspect, it shouldn't be any difference between the use of regular compression stockings or their lack. Answer: all patients had eccentric padding along the treated vein initially. This will be added in the text. We decided for regular compression stockings in our outpatients to make sure that the patients can use the compression devices themselves in a standardized way. This would not have been possible with compression bandages.
  3. Since the study was performed almost 8-9 years ago, it would be exciting to know its results from longer follow-up, especially the long-term results in patients with large diameter veins (>15 mm). Authors should at least shortly comment that issue. Answer: We agree that long term follow-up would be interesting concerning the results concerning long-term occlusion rates and recurrent varicose veins. We added a sentence in the text. However occlusion rates where not the primary endpoint in this paper but pain and discomfort after the procedure.
  4. Which of these measurements was used for comparison? From practical point of view, it is very important for the outcome (especially, the risk of recanalization), whether the vein diameter reaches 21 mm at sapheno-femoral junction, or on the level of middle thigh. Furthermore, in the table 1. the length of treated vein in some patients is less than 50 cm (from 30 to 88 cm). What about the interpretation of the assessment results on the level of 50 cm from SFJ in those cases? Authors should shortly comment that issue. Answer: In Table 1 the reported diameter of the GSV was measured at 3 cm. We add this in the Table 1. During follow-up the complete length of the GSV was checked for occlusion or recanalisation as mentioned in "Methods". The locations at 3, 25 and 50 cm where routinely investigated more intensely according our standard protocol. As this is of no importance in this paper we skipped this part of the sentence in the text.
  5. Methods (line 86): The normal ELVeS Radial 2 ring™ fiber is 1.8 mm, whereas the thinner one (Slim) is 1.2 mm in diameter. Please, verify, since to my best knowledge there is no 600 μm fiber available in Biolitec company. Answer: to our knowledge 1.85 mm is the diameter of the radial tip of the fiber, the fiber itself has a diameter of 600 μm which is important for the energy transfer. The slim fiber is smaller.
  6. Two final sentences at the end of Abstract and in Conclusions are inconsistent. At first Authors have concluded that "Wearing a compression stocking after EVLA reduces pain within the first week on a significant, but low level." and then "Taking the very low differences in pain levels in account, the difference may not be clinically relevant and post-treatment compression may not be mandatory if no additional phlebectomies or sclerotherapies are performed." What is the rationale to postulate, that EVLA with sclerotherapy does require the use of compression, whereas EVLA alone does not? Authors did not include any auxiliary treatment in their analysis. Moreover, the recommendations for the use of compression after sclerotherapy are similarly ambiguous and lacking the good evidence as for EVLA. Authors should better explain their point of view and possibly should refer to recent papers by Pihlaja et al. - Eur J Vasc Endovasc Surg. 2020 Jan;59(1):73-80; Tan et al. - J Vasc Surg Venous Lymph Disord 2021;9(1):264-274, and also European Guidelines by Rabe et al. from Phlebology 2013. Answer: We agree that post-sclerotherapy compression is an ambiguous question. The intention of this sentence in the text was to limit our results to EVLA procedures without any additional phlebotomies or sclerotherapy. We did not intend to postulate, that EVLA with sclerotherapy does require the use of compression. To make this clear we changed this sentence in the text.
  7. Furthermore, the last sentence in conclusions should be accompanied by addition “in short-term observation”, since no long-term results have been presented. Answer: This was changed in the text